# "The Greatest and Most Important Human Right": Citizenship and Bureaucratic Indifference in Refugee-UNHCR Correspondence

Lamis Abdelaaty

Syracuse University
labdelaa@syr.edu

Wednesday, January 31, 24

## Abstract

**This article examines how refugees advocate for themselves with the UN Refugee Agency (UNHCR), and what responses their communications engender. It analyzes letters sent by refugees in Kenya to UNHCR headquarters in Geneva between 1983 and 1994. The findings underline a disjuncture between refugees' efforts to constitute themselves as political agents, and UNHCR's insistence on viewing them as depoliticized subjects. The refugees perform citizenship vis-à-vis UNHCR, using their shared identity as a basis for collective claims-making and trying to renegotiate their unequal relationship with the international organization. To empower themselves, they adopt the international organization's own refugee rights vocabulary and play off different organizations and layers of UNHCR against each other. UNHCR's responses (or lack thereof) demonstrate the consequences of its insulation and bureaucratization. These insights are especially noteworthy in light of recent progress on meaningful refugee participation in the refugee regime.**

## 1. Introduction

Notwithstanding the dramatic increase in academic work on refugees over the past few decades, there remain gaps in our understanding of the role of these individuals as agents in an international political landscape dominated by the UN Refugee Agency (UNHCR), of their engagement with power and rights through grass-roots organizing, and of responses and resistance by the targets of their advocacy.

This article examines how refugees advocate for themselves with the international institution that is tasked with their protection, UNHCR, and what responses their communications engender. Relying on archival research and qualitative data analysis, it analyzes letters sent by different groups of refugees in Kenya to UNHCR headquarters in Geneva between 1983 and 1994, to examine variation in the format and content of letters and in the responses they received. This period marks the transformation of Kenya into one of the world's top refugee hosts and the site of one of the most enduring protracted refugee situations.

The letters, and UNHCR staff's annotations, shed light on contestation between UNHCR and refugees as a feature of the global refugee regime. This article thus puts into conversation three bodies of literature: citizenship studies, scholarship on refugee petitions, and research on international organizations. The correspondence involves refugees exercising political agency and making rights claims, activities that have been theorized as enacting citizenship (see, for example, Isin 2017). However, the

literature on citizenship has thus far not considered instances where performances of citizenship are aimed at an international organization rather than a state. Further, scholarly work on refugee petitions and letters (such as Gatrell et al. 2021) has largely been unable to engage with the ways in which authorities within the refugee regime respond (or do not) to refugee communications. In examining how the letters were received by UNHCR, this article also contributes to the literature on international organizations. Studies that conceive of international organizations as bureaucracies (e.g., Barnett and Finnemore 2004) are particularly relevant here, though it is worth noting the unique features of this analysis wherein political interactions are taking place directly between individuals and international organizations, without state intermediaries.

The findings underline a struggle over the political agency of refugees: whereas the refugees perform citizenship by making rights claims of UNHCR, the Agency discounts their demands and denies their ability to participate actively in the refugee regime. The refugees do not organize exclusively by national identity (and, indeed, there is evidence of conflict within refugee groups of a single nationality), suggesting that they may be mobilizing as citizens of UNHCR rather than of particular countries. The complaints they make about the Kenyan government and UNHCR's local personnel imply that the Agency not only bears ultimate responsibility for the refugees, but also ought to be accountable to them.

Further, a range of demands (from resettlement to refugee oversight to macro-level reforms) are often couched in terms of refugee rights and references international refugee law. Indeed, the refugees' letters use UNHCR's own language to legitimize their claims and stress their deservingness of rights as "political refugees," to convey the severity and urgency of their conditions, and to elicit the compassion of readers. Given refugees' asymmetrical relationship with UNHCR, they often call in (by copying their letters to) other layers of the organization and external actors.

UNHCR's written commentary on these letters exposes a bureaucracy that is accepting of perceived political constraints, even when this comes at the expense of refugee rights. Most letters from refugees receive no reply, in stark contrast to letters from third parties. Frequently, the correspondence is simply "noted" or else forwarded to the Kenyan branch office. Efforts by a refugee-led organization to gain recognition and to partner with UNHCR are dismissed. In short, UNHCR's insulation and bureaucratization result in a situation where the organization refuses to be held accountable to refugees, or even to recognize them as political agents.

## 2. Refugee Agency, Rights Claims, and Bureaucratic (Non-)Responses

In a hugely influential article, Malkki (1996) described refugees as "speechless emissaries" of suffering. Bureaucratized humanitarian practices depoliticize refugees, Malkki argued, rendering them "pure victims" and denying them "the authority to give credible narrative evidence or testimony about their own condition in political and institutionally consequential forums" (378). As subsequent studies have noted, there then arises a tension between dominant representations of helpless refugees and refugees' own political activities (see, e.g., Sigona 2014).

This article contributes to the growing literature on refugee agency (Chatterji 2013, Gatrell 2013, White 2017, Nowak 2019b). For example, Moulin and Nyers (2007) document how Sudanese refugees in Egypt, having been denied recognition as political actors, used protest to make demands of UNHCR. This article is similarly interested in refugee collaboration and organizing at the grass-roots level in order to exercise influence, advance common interests, and advocate for themselves in the face of UNHCR's hegemony. In centering refugees' political agency, it further emphasizes the role of refugees as active participants in the global refugee regime who seek to shape and reform it. Moreover, it provides insight into refugees' views of international politics and refugee rights, and how they see the power imbalance between themselves and UNHCR.

Addressed to the High Commissioner, the letters this article examines represent an instance of "writing upwards" (Lyons 2015). Like other examples of written communication between two correspondents who are very socially and politically unequal, these letters attempt to circumvent bureaucratic procedures via a personal approach. Through their letters, the refugees try to bridge the distance between their experiences on the ground in Kenya and their interlocuters in UNHCR's Geneva headquarters, cutting out intermediaries and seeking direct communication with an international institution. For leverage, the refugees try to play off different organizations and layers of UNHCR against each other.

My analysis demonstrates, however, that refugees do not behave like the "supplicants" or "citizens" in Fitzpatrick's (1996) dichotomy: their writing attributes a benevolence to UNHCR, but they are nonetheless forceful in stating their criticisms and suggestions. Rather, the letters may be fruitfully viewed through Isin's (2017) lens of performative citizenship. Isin notes that citizenship can have a transnational/international character and may be performed by people who are not citizens in the conventional sense, so long as they are acting as political agents by making (or claiming the right to make) rights claims. Indeed, the letters reveal refugees engaging in both everyday (or quotidian) performances of citizenship and also spectacular acts (such as a hunger strike). In other words, these are acts of citizenship in the absence of legal citizenship status (Swerts 2017).

Cooper (2016) conceives of citizenship as constituted by equivalent membership in a collectivity. Indeed, the specific lines along which refugees organize to write letters not only suggest that they view themselves as a community, but signals an equivalence among them that transcends national identity and other divides. Similar to the process whereby enacting citizenship involves actors differentiating themselves from "strangers, outsiders, aliens" (Isin 2008), the refugees' letters define their belonging and membership in opposition to economic migrants.

What, then, is the "city" of which refugees are claiming political membership? Nearly all studies on citizenship focus on state authorities as the target of rights claims. In this case, however, refugees are performing citizenship vis-à-vis an international organization, not a government. The literature on international organizations generally views states as intermediaries between their citizens and these bodies. But refugees cannot rely on their home governments, responsible as these authorities are for their displacement in the first place, to act in this capacity. Without a "right to have rights" (Arendt 1951) upheld by their home state, refugees reach out to UNHCR directly. According to a "participation" model of accountability, power-wielders should be accountable to those who are affected by their actions (Grant and Keohane 2005). Indeed, Harley (2020) demonstrates that refugees were once influential in the development of international refugee law and policy, during the period 1921-1955. Through their letters, refugees once again seek to establish themselves as agentic constituents of the refugee regime, constituents who are entitled to hold UNHCR (a cornerstone of that regime) accountable.

UNHCR is, after all, tasked with safeguarding refugee rights and has the moral and expert authority to shape the content of these rights. Moreover, around the world, UNHCR functions as a "surrogate state," registering and documenting refugees, providing food and shelter, administering social services, managing camps the size of small cities, and establishing policing and justice mechanisms (Kagan 2011, see also Micinski 2022; Moschopoulos 2023).[1] Bender (2021, 13) argues that UNHCR is the "de facto sovereign" of refugee camps, wielding "direct executive power" over refugees (see also Agier 2011). Just like a state, UNHCR can be said to have "its own territory (refugee camps), citizens (refugees), public services (education, health care, water, sanitation, etc.), and even ideology (community participation, gender equality)" (Slaughter and Crisp 2008, 8). No wonder, then, that multiple refugees have said "we live in a country of UNHCR" (Grabska 2008, 87).

Rights claims that target state officials represent "the recognition of state authority, the acceptance of subjects being governed" (Leonardi & Vaughan 2016, 96). Here too, refugees' letters concede that UNHCR wields political rule over them. Nonetheless, refugees are attempting to have a say in how the Agency's power is exercised. Acting as citizens is a "strategy of interruption" into an order in which individuals are seen as passive victims (Nyers 2008). In refusing to be silent and disempow-

ered, refugees challenge dominant representations and UNHCR's expectations. Expressing grievances and making demands, the letters show that refugees seek to transform themselves from depoliticized subjects into actors and to gain recognition of both their agency and their right to protection.

These acts of citizenship are an attempt to redefine refugees' relationship with UNHCR, to cast refugees as the constituents whom UNHCR serves and to whom it ought to be accountable. To legitimate their claims, the refugees adopt the language of international law and the refugee regime – UNHCR's own vocabulary. They appeal to human rights and international norms, to UNHCR's own responsibilities to refugee populations, and to the texts of international treaties. Implicitly, the letters call into question UNHCR's legitimacy as the Agency that speaks for refugees.

There is, of course, a burgeoning scholarship on refugee petitions and letters sent to authorities within the refugee regime (Nowak 2019a, Irfan 2020, Gatrell et al. 2021), which itself is part of a scholarly debate on refugee perspectives and "voices" (see, e.g., Butalia 1998, Ranger 2005, Kindersley 2015). My analysis goes further, however, by engaging with UNHCR responses to refugees' letters. Since citizenship is fundamentally about the relationship between "units of belonging" and "units of power," we must examine how rights claims are resisted (Cooper 2016, 286). Indeed, refugees' letters, which challenge UNHCR's preference for obedient (and silent) refugees that do not challenge its authority, are often met with refusal and denial.

UNHCR's responses (or lack thereof) are indicative of the pathologies that are revealed when international organizations are analyzed as bureaucracies (Barnett and Finnemore 2004, 39-40). My analysis indicates how the embeddedness of rules and procedures can impede the functioning of the organization: staunch commitment to the chain of command and routing procedures lead to letters that express grievances about branch offices being forwarded to those same locations. Bureaucratic universalism flattens important diversity: most appeals receive similar or identical responses despite their unique characteristics. Finally, insulation results in a situation in which refugees must resort to direct appeals to try (and often fail) to impose themselves on a distant UNHCR headquarters. Indeed, UNHCR's organizational culture may encourage dismissive and patronizing responses, fueled by an underlying assumption that refugees are dependent on UNHCR because of an inability to help themselves (Wigley 2005, 33).

Whereas refugees may be performing citizenship by making rights claims, UNHCR regards refugees as subjects – or perhaps "protection objects," to use Krause's (2021) phrase. UNHCR may have to answer to states who have delegated authority to it (Abdelaaty 2021), but it does not see itself as accountable to refugees who are vulnerable to its policies (Barnett and Finnemore 2012). In place of democratic accountability, the organization rules its constituents via "compassionate authoritarianism:" it seeks to relieve refugees' suffering while denying them access to grievance procedures (Holzer 2015). Put differently, UNHCR's paternalism fuses care with control (Barnett 2011). Refugees view themselves as citizens and UNHCR as a polity, but that does not align with UNHCR's views of the refugees or of itself. The correspondence analyzed in this article is thus the site of political struggle between an organization that has the power to shape refugees' fates, and refugees who are attempting to hold it accountable to its own principles.

## 3.  Letters from Refugees in UNHCR Archives

The analysis in this article relies on documents collected from the UNHCR Archives in Geneva, Switzerland. Specifically, I focused on Series 1 to 3 (Classified Subject Files) of Fonds 11 (Records of the Central Registry), which detail the Agency's activities in countries around the world between 1951 and 1994. Within these, I examined all folders relating to Kenya with the headings: External Relations, Specific Refugee Situations, Eligibility, Accreditation, and Administration and Finance. These folders primarily contain internal UNHCR communications (e.g., between headquarters and branch offices, or between different units at headquarters) as well as memos, notes for the file/meeting minutes, and the occasional field mission or situation report. The documents also include correspondence with government authorities, NGOs, and other UN bodies.

Documents within the folders are separated into folios organized by date. Letters from refugees are not classified separately and were interspersed among these documents; my search uncovered 35 folios (a total of 346 pages) that each included letters (35 in total) penned by various groups of refugees including Ethiopians, Ugandans, Somalis, and others between 1983 and 1994.[2] Some of these letters are handwritten, others are typed. A few were sent via fax, but most were mailed – in some cases the stamped envelope was archived as well. Each folio is accompanied by an "action sheet" which marked its processing by UNHCR's staff (see Figure 1). These printed action sheets were filled out by hand. They include information on when the document was received, who was designated to take action (the "action officer"), which individuals it was subsequently routed to, what queries or comments they had, and whether a reply was sent (and if so, its reference number and date).

**Figure 1. Example of UNHCR Action Sheet**

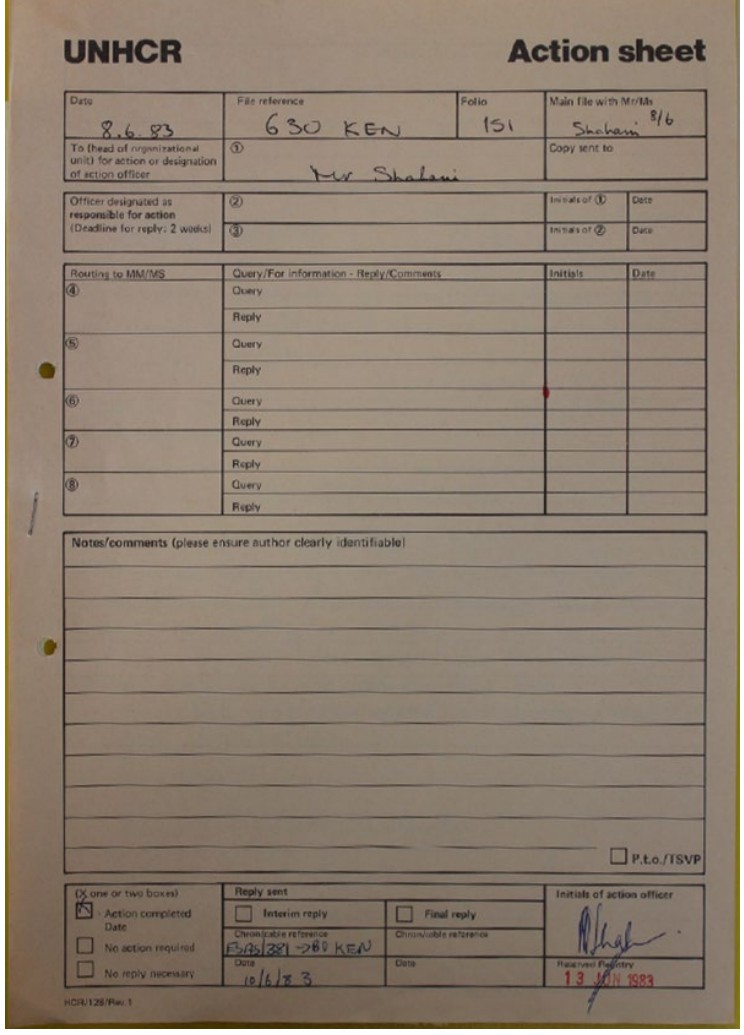

Action Sheet, 8 June 1983, *UNHCR Archives*, Fonds 11, Series 2, Box 1198, 630 KEN Protection and General Legal Matters - Eligibility - Kenya [Volume 2], Folio 151.

Using the qualitative data analysis software QDAMiner, I coded text segments in each letter following an inductive approach to "identify the dimensions or themes that seem meaningful to the producers of each message" (Abrahamson 1983, 286). These dimensions include how the authors/signatories identified themselves, the displacement narrative they present, the complaints listed, the requests issued, and any justifications advanced. The text of action sheets and inter-branch cables was also coded to reflect which UNHCR officials were assigned responsibility for each letter, what actions they recommended, and how they justified these actions. I also noted any words or phrases that were

emphasized (i.e., underlined or capitalized) and the overall tenor of each document alongside other characteristics. A fuller list of codes can be found in the Appendix. This methodologically rigorous approach enabled me to uncover trends and patterns in the letters and associated documentation, including how frequently the codes occur and which codes tend to appear concurrently.

The letters that I analyze are certainly not representative of refugees' activities and UNHCR's responses across time and space, but they are nonetheless indicative. The following section begins by providing contextual information about UNHCR and refugees in Kenya before presenting my analysis of the letters. I quote extensively from these documents so that readers can grasp how refugees expressed their concerns in their own words.

## 4.     Missives from Kenya, Processing in Geneva

Though Kenya has received refugees since its independence in 1963, these were generally few and only numbered 6,760 (predominantly Ugandans, Ethiopians, and Rwandans) in 1983 (UNHCR n.d.). Then, with increasing arrivals of Somalis and Sudanese, Kenya's refugee population jumped from about 14,000 in 1990 to over 400,000 in 1992 (UNHCR n.d.). At the end of 1994, Kenya hosted 252,423 refugees: Somalis represented the largest share at 82%, followed by Sudanese refugees at 11%, Ethiopians at 4%, and smaller numbers of Ugandans, Rwandans, and others (UNHCR n.d.).

The Kenyan government retained responsibility for refugees until 1991. With the mass influx of refugees in the early 1990s, responsibility for refugee status determination was transferred to UNHCR. Delegation to UNHCR was also accompanied by a decision to require refugees to reside in camps. UNHCR went along with the government's encampment policy by undertaking camp management. NGOs, as implementing partners, were subcontracted to provide social, health, and community services in the camps (Veney 2007).

Meanwhile, it was a time of change at UNHCR headquarters as well. Starting in the mid-1980s, High Commissioner Jean-Pierre Hocké elevated the Agency's material assistance operations while downgrading its previously dominant legal protection arm. Concurrently, many within the Agency began to view refugee repatriation as the most realistic solution for refugees, even if circumstances in their home country were less than ideal (Betts, Loescher, and Milner 2013). This was also a time of fluctuating financial resources for the Agency, resulting in significant budgetary shortfalls (Loescher 2001).

In this context, refugees in Kenya penned the letters described below. The following pages present an analysis of the letters centered around how refugees organized, how they described their displacement, which grievances they expressed and demands they made, what strategies they employed for persuasion and leverage, and what UNHCR officials had to say in response. This sequence generally follows the structure of the letters themselves, which usually begin with writers identifying themselves and then unfold with a description of their journey, a list of complaints and requests, and finally reasons why UNHCR ought to be receptive to their message. As will be shown below, each narrative step can be viewed as constituting performative citizenship.

### 4.1. Identities

"By crossing the border, we became a UNHCR statistic, to wit asylum-seekers, refugees," explained a Somali man in Kenya when interviewed by Farah (2000, 16). The implication is that individuals' many particularities are disregarded – the refugee label eclipses all other aspects of their identity (Horst 2007, 13-14). However, as Zetter (1991, 55) argues, refugees may not be "unwilling victims" here; their politicized identity and "enhanced solidarity may be turned to advantage as a lever on governments and agencies." In line with this contention, the letters I analyze show refugees' embracing a shared refugee identity that allows them to perform citizenship.[3] This equivalent membership in a collectivity resembles the notion of "campzenship" proposed by Sigona (2015) to capture the ways a camp can produce situated ways of being political, though the letters described below are not exclusively from refugees residing in camps.

UNHCR tended to organize the correspondence it received by refugees' country of origin, with separate categories for "Ethiopian Refugees in Kenya," "Somalian Refugees in Kenya," and so on. Refugees did sometimes identify themselves in this way in writing, particularly when there were no other nationalities housed in their specific camp or settlement.[4] However, the refugees did not organize exclusively based on national identity. Rather, the letters demonstrate that refugees sometimes banded together in subgroups according to shared characteristics, including ethnicity (e.g., Ugandan Nubians[5]). In doing so, they may have been able to make claims specific to their identity, as when a group of Somali women pleaded for help for their "poor futureless children, who half of their fathers were killed in civil war."[6]

For example, committees composed of former government officials who escaped Ethiopia following the fall of the Mengistu regime in May 1991 described their unique situation and concerns. In one letter, the "Ethiopian Refugee Committee" composed of individuals who had "served in high civil, military and se-curity posts under the previous government" expressed alarm at the Nairobi assassination of Oromo liber-ation leader Jatani Ali by Ethiopian security agents and called for increased attention to their safety.[7] In a subsequent missive, 41 former members of the Ethiopian Ministry of Internal Affairs complained that they had long been deprived from assistance due to their political background: "However very meratori-ous [*sic*] our cases are, we are approaching to two damn solid years without enough attention paid to us."[8] Similarly, the "Committee for Ethiopian Civil & Military Officials" asked whether they were being de-nied access to protection because of their former positions.[9] In a follow up letter, the chairman of the now renamed "Committee for Ethiopian Political Asylum Seekers" asked how resettlement opportunities could be denied to lower-ranking officials given that top officials of the former Ethiopian government had been granted asylum in Western countries.[10]

In other cases, refugees with shared interests organized across nationality groups, as with the "Disabled Committee" at Marafa representing Ethiopian, Somali, and Sudanese refugees with physical and mental disabilities.[11] A letter from Thika Reception Center brought together Ethiopian, Rwandan, Somali, South African, Sudanese, Ugandan, and Zairean refugees denouncing their mistreatment by Kenyan staff and complaining that "the UNHCR agency either by intention or incompetence or corruption does not care."[12] When it was announced in March 1992 that refugees would be transferred from Thika (which is close to Nairobi) to remote camps in Ifo, Liboi, and Walda, a "Joint Ethio-Somali Committee" wrote to the High Commissioner to ask that UNHCR and the Kenyan government reconsider; among other reasons, they noted that both Somalis and Ethiopians would be vulnerable to cross-border attacks, and that the camps were already overcrowded and lacked even basic supplies.[13]

## 4.2. Displacement Narratives

Some letter writers described their journeys to Kenya as a preamble of sorts. Notably, across these diverse groups, they were often careful to emphasize that they were "political" refugees. This insist-ence demonstrates a keen understanding of the "refugee/migrant binary" (Hamlin 2021), which casts some people as deserving refugees who were forced to flee for political reasons, and others as undeserving migrants who voluntarily move for economic motives. Designating themselves as refu-gees *and not migrants* signals their deservingness of UNHCR's help, and also defines their citizenship. They are members of a political community that transcends national identity, but that nonetheless excludes non-refugees (who do not belong).

Economic conditions in refugees' countries of origin were almost never mentioned as a trigger for displacement.[14] In fact, some letters appear to invoke the binary directly: "almost all of the refugees at Thika were forced to flee their respective countries not due to the economies, but is due to the politics of their governments."[15] Similarly, a letter from Ethiopians at Walda clarifies "we refugees fled from our homeland not because we are famine striken [*sic*] and starved." Declaring their oppo-sition to the Ethiopian People's Revolutionary Democratic Front (EPRDF), the letter exclaims "WE ARE POLITICAL ASYLUM SEEKERS!" [emphasis in original].[16]

Even when generalized violence was mentioned, it was almost always in conjunction with a discus-sion of targeted persecution and human rights violations.[17] In May 1983, a group of Ugandan Nubians wrote from Busia (a town on the Kenyan-Ugandan border) with a detailed account of their displace-ment from Uganda, which began with the 1979 Liberation War:

Our reason of taking refuge into kenya [*sic*] was due to the mistreatment we receive during the liberation war. Foremost of all, we were denied our rights of employment and businesses, our houses were all broken down and our properties totally looted. Our people were detained in prisons without trial, others died in the prisons and some killed during the war of which a list of few of those killed and still being detained will be attached. In addition, our accounts in different banks in Uganda were frozen until today.[18]

After the fall of Idi Amin, the letter states, Kenya returned the Nubians to Uganda where they were imprisoned by the Uganda National Liberation Front (UNLF) for alleged crimes committed during the period of military rule. During this time, the letter says, "we lost so many children and few adults." Upon their eventual release, they were "asked to go back to their homes but unfortunately they were welcomed by beatings and killings so most were killed during this time. To speak the truth others were again re-arrested and taken for detention in prisons."[19]

Meanwhile, "Our people who have risked their lives to stay back in Uganda have suffered and are still suffering. The few property they owned were again looted and they received beatings now and then from the undisciplined soldiers of Uganda." The letter is penned by "We who found that life in Uganda was impossible and it was unworthy living. We decided to flee once again into kenya [sic]." The letter concludes with a list of 87 names of Nubians "who were killed and those who died in different prisons in Uganda as a result of the liberation war."[20]

## 4.3. Grievances

In describing Somali refugees in Kenya, Hyndman (2000, 156) notes that they "have a reputation of talking back to relief workers, rejecting the charity script of the needy *and* grateful;" their actions "unsettle the charitable, hierarchical relationship of power between Western donors and Somali refugees." The letters analyzed here also contain evidence of refugees negotiating their relationship with UNHCR, contesting the Agency's control while still recognizing its authority. Most of all, they seek to construe themselves as citizens, and not subjects.

Despite presenting grievances and, at times, betraying refugees' frustration and indignation, nearly all of the letters adopted a formal and deferential tone. Further, many letters begin by thanking the Kenyan government and/or UNHCR, to provide one example: "we wish to take this opportunity to deliver our heartfelt gratitude to the President and the government of the Republic of Kenya and to the UNHCR. We greatly appreciate the efforts made to save the lives and accommodate the refugees and the hospitality, with all means and capacities at your disposal."[21]

The specific complaints raised in the letters are varied. Some letters appealed for assistance with meeting refugees' basic needs. For example, Elders in Utange denounced "the unfavourable life conditions prevailing in this camp" in a letter addressed to UNHCR's Nairobi branch office and cc'ed to UNHCR headquarters. Given the poor quality of maise flour, hard-to-cook beans, and limited amount of sugar in refugees' daily rations, the camp's 10,000 inhabitants were "facing spectre of starvation." With no cooking charcoal, the Elders feared that refugees would have to cut trees "with tremendous environmental damage," or else burn the wood from which their shelters were constructed. Moreover, the water supply failed frequently and a planned water tank had not been built. Somali health care professionals were providing care, but received no renumeration and risked running out of medicine; several refugees had already died from malaria and other diseases such that "the camp may become the graveyard of the Refugees instead a place of asylum for them." Finally, hundreds of children and young people "elementary 460, intermediate 272, secondary 196, university 220, technical schools, etc" were receiving no education whatsoever.[22]

Still, in other correspondence, refugees objected to being placed in a situation of forced dependency in camps instead of being able to earn their own livelihoods. For example, Ethiopian refugees at Kakuma emphasized that their community included highly educated individuals and experienced professionals, noting "we feel that we are dehumanized when we see only food being distributed to us as if we were cripled [*sic*] and could not pursue A Better Path of Life." They expressed confidence that

they could "contribute to the betterment of the society … if we are given the chance to pursue our life in a relatively better place than this refugee camp."[23]

Others urged protection from security threats. A letter from "Representatives of the Multi-Ethnic Ethiopian Refugees' Community at Walda Refugee Camp" recounts that they were accustomed to discussing their situation with the UNHCR field office and police stations but decided to reach out to UNHCR headquarters after conditions worsened:

> … nowadays, it has become a common phenomenon to hear bullets showering over Walda. As a result, people were shot dead, roasted by fire and slaughtered like animals. To be specific, a score of people were killed and robbed of everything they possessed in September and on December 6, '92. And three bodies were found lying in the forest. Unfortunately, little is known of who did this. Rumours are, however, spreading that this is the extension of the ethnic conflict between Borenas and Geris in Ethiopia. It is said that these two Ethiopian tribes are here escaping from perhaps the fight and/or the drought. Actually, members of either tribe were repeatedly seen here in the camp running against those of the other tribe. In spite of our position on this matter being neutrality, the effect might directly or indirectly touch us since all of us share the same country and bear same nationality. We do not support actions of either of the tribes. And yet we feel that we are sandwiched between them, and that we ought to pay for it. For instance, one of our community members was wounded in the crossfire on September 30, '92.[24]

The letter also notes that an individual captured at Walda police station had admitted that he was a spy sent by the EPRDF to gather information about Ethiopian refugees. The representatives requested that their community be moved to another camp "where we can live in good harmony with each other, regardless of our nationalities or tribes."[25]

One letter suggests a conflict between refugees. A group at Utange wrote to object to the Elders Committee proposed by some refugees and recognized by UNHCR's sub-office in Mombasa, proclaiming that it was "totally against the will of the majority and can't function in this camp; because it is absolutely formed for the subjugation and putting pressure on our refugee rights, and the welfare of our people in the camp and we can't accept to kill our people with our hands."[26] The authors insisted that a new committee be formed that reflected the numerical distribution of different tribes.

Occasionally, the complaints raised related to the Kenyan government, objecting for example to government orders for them to relocate to another camp. A later letter from Somali Elders in Utange camp recounts how they "invested in it physical, mental and financial resources" over 14 months to make it livable, "establishing different social services in the Camp such as school for our children, Health services, a Social hall, Worship places, Playing grounds, Tea shops, etc." Somalis were in "harmonious cooperation" with the local Giriyama ethnic group and the population of nearby Mombasa, and "no relevant security problems involving them have taken place." If Kenyan authorities were intent on closing Utange, then the refugees preferred either repatriation to secure regions in Somalia or asylum in a neighboring country.[27]

Far more commonly, refugees' grievances centered around UNHCR's staff in Kenya. Some complaints alleged negligence, like a letter addressed to Under-Secretary of the Ministry of Home Affairs and cc'ed to UNHCR's Geneva offices which complains that UNHCR's branch office in Nairobi (along with its implementing partners) "have turned deaf ears on us" when it came to assistance and education.[28] Accordingly, some letters indicate that refugees decided to contact headquarters after their complaints went unaddressed by the branch office.

Indeed, several letters contend that UNHCR staff in Kenya were not only incompetent but complicit in refugees' plight. One letter notes that hundreds of Ugandan refugees had been forcibly repatriated "with the full knowledge of UNHCR. We therefore request or ask Geneva or UNHCR Headquarters to come to our rescue since the UNHCR Kenya seems to be part and parcel of our suffering … Otherwise we shall be forced to take or interprait [*sic*] the organization a political organ but not a humanitarian or none political has [*sic*] its supposed to be."[29]

One letter even alleged corruption and exploitation at UNHCR's Nairobi branch office. A group of Ethiopian refugees at Thika wrote that nothing had been done to assist them with repatriation or resettlement. Not only did UNHCR's protection officer repeatedly miss appointments (forcing refugees to incur significant transportation expenses in order to return), but refugees had to pay bribes in order to enter UNHCR's compound and to purchase each asylum-seeker form for KSh50 (these are meant to be distributed freely). "We are refugees and not TOURISTS," the letter continues. At Thika Reception Center, staff were selling relief items on the market instead of distributing them to refugees. At the center's clinic, the nurse distributed aspirin to treat malaria and intestinal diseases. The letter concludes that "the UNHCR (Kenya), and the refugee camp of THIKA can be called 'MONEY MAKING INDUSTRY, UNLIMITED SHARE COMPANY' to fill the pockets of its officials."[30]

## 4.4. Demands

Cooper (2016, 287) notes that citizenship "enables people to push on the form, as well as the actions, of the state." The requests made in the letters at times concerned themselves only with UNHCR's actions, but at other times went much further to demand substantial refugee involvement and participation in the refugee regime. In Cooper's terms, these letters therefore demonstrate a form of "thick" citizenship that does not claim formal membership, but rather demands protection from abuses, makes rights claims, and insists on collective action.

Worth emphasizing is that the letters do not reveal a uniform desire for resettlement among the refugees. To be sure, some individuals and groups appealed for resettlement, noting that they were denied the possibility of local integration and could not be safely repatriated.[31] Others requested to be repatriated, however. Some refugees appeared to have accepted their long-term encampment and only requested they be relocated (or not relocated) to another camp. Several letters simply pleaded for material assistance with food, shelter, medical care, educational needs, and other essentials (e.g., clothing).

More striking are the letters that go beyond these requests. For example, the aforementioned letter from Ethiopian "political asylum-seekers" at Walda concludes with a list of "demands," which include: that their status determination cases be processed by UNHCR (following a delay of more than four months), that they meet with counsellors and other officials "at least every fortnight," and that a "permanently stationed Protection Officer" be assigned to their camp.[32] Meanwhile, the Ethiopian refugees at Thika who, as previously described, alleged corruption insisted that Geneva send a fact-finding mission to investigate UNHCR's Nairobi branch office. In addition, they asked that the editor or a staff member of *Refugees* (a magazine published by UNHCR) be sent to them so that they could "AIR OUR PROBLEMS" [emphasis in original].[33] Finally, in the letter described above detailing the lack of basic supplies and services at Utange, the Elders recommended that refugee representatives be permitted to exercise oversight or "followup of all supplies meant for them."[34]

Remarkably, one letter called for "big picture" reforms, including refugee representation at the UN, that assert the role of refugees as international actors and reveal knowledge of international politics and intergovernmental organizations. The impetus for this letter, addressed to the UN Secretary-General and cc'ed to the High Commissioner for Refugees, appears to be a directive issued by President Moi ordering all Ugandan and Rwandan refugees to leave the country (Africa Watch 1990). The letter notes that refugees had been given one week to leave Kenya, but they were unable to repatriate: "It is impossible for us refugees living here to drink poison, to hang ourselves with ropes, to cut ourselves with knives, to kill ourselves, our children and our grand children and so save the world the curse of being refugees."[35]

A numbered list of requests includes that refugees in Kenya should be represented in the UN Security Council and other UN bodies. In addition, refugees living in Kenya should be assisted in establishing "their own international refugee organisation (IRO) of, for, and by refugees of this world themselves – through which and in which all refugees of this world can and may freely, openly, lawfully, peacefully, fully, permanently and effectively defend, protect and preserve all their human rights." The final demand is that all refugees living in Kenya be supported "to convene and hold immediately an EMERGENCY INTERNATIONAL REFUGEE CONFERENCE." This conference would consider how to resolve

the refugee problem "once and for all," make recommendations to the UN Security Council, and "give IMMEDIATE assistance, aid, help, support etc of all kinds to all refugees of this world."[36]

The letter is titled "Human Rights of Refugees" and repeatedly refers to the "greatest and most important" human right for refugees. This is not the right to asylum or non-refoulement, but rather that "refugees of like minds and like hearts can and may freely, openly, lawfully, peacefully fully organize and unite themselves in their own national, continental, international, political and non-political organizations for and by refugees of like minds and like hearts alone."

Further, it does not specify the nationality of its authors, advocating on multiple occasions for "ALL refugees of this world." It cites examples like those of "the Romanian refugees in Austria, the Vietnamese refugees in Hong Kong, the Burmese refugees in Thailand, the Liberian Refugees Sierra-Leone." It concludes with 21 signatures, "For and on behalf of all the terribly suffering, poor, despised, oppressed, humiliated, and dehumanized refugees living here." [37]

## 4.5. Argumentation, Evidence, and Leverage

The strategies used by refugee letter writers to legitimate their claims also connect to citizenship. As Horst (2007) notes, the refugee label "entails the entitlement to certain rights." Refugees may claim these rights by emphasizing their vulnerability or their oppression, demanding assistance and possibly also recognition from other international actors. Indeed, many of the letters described above underscore refugees' devastating conditions ("We are … treated like the victims in Nazi concentration camps" [38]) and the urgent nature of their queries ("Please come to our aid, before we all die like unwanted animals"[39]). They appeal to compassion ("prevail justice on our side"[40]) and the rectitude of their readers ("Please, East-West, North-South without any geographic barrier, ideological differences, color discrimination, save us, watch us."[41]). In this way, the letters bring to mind Alexopoulos's (1997) "ritual lament," an emotional account of a pitiful life that identifies the reader as the only agent capable of reversing the author's misfortune, thereby challenging them to act.

In some cases, letters were accompanied with evidence to substantiate their claims. As noted above, one letter concludes with a list of names of individuals killed to demonstrate the dangers members of this group escaped.[42] Another letter recounts the capsizing of a boat carrying 640 Somalis near the town of Malindi in February 1991. In the letter, a group of seven Somali elders at Utange (who had survived the incident) complain that Kenyan police officers arrested those who attempted to rescue the boat's passengers, while some local residents stole Somalis' belongings. Before demanding an investigation and compensation, the letter provides a list titled "Source of Inforrmations." UNHCR headquarters are instructed to contact two witnesses (hotel owners in Malindi) and to consult a news article and an ICRC-Kenya report for corroboration and additional evidence.[43]

Notably, the letters often legitimated their claims by making reference to refugee rights under international law and UNHCR's own regulations. For example, in relation to its complaint that low-ranking former officials were being denied resettlement, the "Committee for Ethiopian Political Asylum Seekers" quoted the full text of the Universal Declaration of Human Rights' Article 2 (non-discrimination) and Article 14.1 (right to seek asylum).[44] Another letter refers to UNHCR regulations in order to decry the unrelenting pressure from its personnel to repatriate.[45] A number of other messages refer to the 1951 Refugee Convention and its 1967 Protocol.[46] Elsewhere, a group of refugees reminded readers that "the main functions of UNHCR are to protect refugees and to seek durable solutions to their problems." [47]

Refugees were also willing to press forcefully for their demands, as shown in a series of letters from a self-identified representative for Ethiopian refugees. In the first letter dated 11 June 1987, the author claimed that UNHCR Nairobi harbored a "hatred" for Ethiopian refugees. Ethiopian asylum-seekers who had seen their refugee status applications rejected by UNHCR had lost their lives attempting to flee to other countries. In light of this, the representative insisted, nine asylum-seekers who had been recently rejected would nonetheless remain at Thika.[48] A follow up letter penned on 17 June complained that food supplies had been cut off to Thika, and Ethiopians were now being asked to leave. Hoping to receive a response, they had decided to initiate a hunger strike starting 20 June.[49]

The third and final letter, from July 1, explains that they had been forced to end the hunger strike due to an attack by uniformed "KENYAN GUN-MEN" [emphasis in original] (presumably police forces) who arrested the rejected asylum-seekers and blocked members of the press. There had still been no reply from UNHCR headquarters in Geneva, but the author demanded they "look after our lives in danger."[50]

For leverage, refugees copied their correspondence to different layers of UNHCR and government officials. For example, a complaint that Ugandans in Thika and Nairobi were not being interviewed by UNHCR's Nairobi branch office was addressed to the branch office and cc'ed to UNHCR headquarters.[51] Another letter, addressed to the Minister for Home Affairs and cc'ed to two UNHCR Geneva officials, claimed that a staffer who had rotated into a position at UNHCR's Nairobi branch office was "too racist and naturally hated black people whether refugees or nationals." She was depriving refugees in order to decrease spending and "get a name for herself as a good worker." The letter called on the Minister to expel the staffer and "from now on scrutinize the UNHCR staff before they start working in Kenya."[52]

Refugees also frequently cc'ed their letters to outside actors, in what appeared to be an effort to exercise pressure on UNHCR. A group of Ethiopian refugees wrote to decry the decision by UNHCR's Nairobi branch office to place refugees who had fled Mengistu's regime in the same camp with officials from his government who were now escaping following his ouster. Their letter is addressed to UNHCR headquarters and cc'ed to the Nairobi branch office, Amnesty International, the European Community, the Canadian embassy, and the Kenyan Ministry of Home Affairs.[53] Similarly, complaints by Ethiopian refugees at Marsabit included that, despite waiting for seven months, they had received neither screening nor recognition of refugee status.[54] Their letter was sent to UNHCR headquarters and the branch office in Nairobi along with government officials (the Kenyan Ministry of Home Affairs and the Marsabit District Commissioner), international NGOs (Amnesty International, International Committee of the Red Cross), an intergovernmental organization (UN-Habitat), a religious group (the Catholic Church), foreign governments (the US Embassy, the British High Commission, the High Commission of Canada, the Australian High Commission), media outlets (Voice of America, the Daily Nation, the Kenya Times), and others.[55]

In one case, a group of refugees issued a press release. In June 1992, one of the leaders of FORD (Forum for the Restoration of Democracy, a Kenyan political party) alleged that Somali refugees were registering to vote in Mombasa and threatened to burn down Utange camp. In response, representatives of the Somali refugee community at Utange wrote a press statement which they shared with UNHCR's sub-Office in Mombasa and three newspapers (the Standard, the Daily Nation, and the Kenya Times). The document denied the allegations and emphasized that the refugees were "innocent, against and extremely sensitive to be dragged to Kenyan political affairs." The statement appealed to Kenyans and the international community for protection.[56]

## 4.6. UNHCR's (Non-)Responses

In Hyndman's (2000) framework, refugee "subcitizens" who lack legal status in Kenya are administered by UNHCR's "supracitizens," international professionals many of whom carry UN diplomatic passports. These two groups, refugees on the one hand and UNHCR personnel on the other, are "ranked hierarchically on the basis of citizenship" (111). When the letters described above were received at UNHCR headquarters, the power differential between refugees and the Agency comes into even sharper relief leading to denial and dismissal of refugees' rights claims.

The "action sheets" which accompany each folio indicate that most (if not all) letters were routed to UNHCR's Regional Bureau for Africa (RBA). Previous research suggests that staff in UNHCR's regional bureaus prided themselves on a "pragmatic" approach that acknowledged political pressures and operational constraints, compared to protection staff who had a strictly legalistic and rights-based orientation (Barnett and Finnemore 2012, 96-97). This bureaucratic culture may account for a willingness to accept prevailing conditions as unavoidable consequences of government policies that UNHCR cannot (or should not) attempt to challenge. In addition, as I have argued elsewhere (Abdelaaty 2021), UNHCR's desire to safeguard its continued in-country presence makes it sensitive to government sanction and constrains its activities.

Officials' comments on some of the action sheets are in line with these expectations. For example, when the "Joint Ethio-Somali Committee" protested their transfer from Thika to remote camps, citing insecurity and lack of supplies in these camps, an official acknowledged "There are definitely some valid points in this appeal," but noted that refugees who chose to move to Uganda would nonetheless receive no UNHCR assistance and would be encouraged to return to Kenya.[57] A fax to the Nairobi branch office instructed them to reply to the letter, remarking somewhat patronizingly "you may wish to indicate to the joint committee the problems they may create for themselves if they move to other countries (first country of asylum principle) and that UNHCR in these countries in general is not ready to give assistance to these people."[58]

Similarly, the first written reaction to the letter from the "Committee for Ethiopian Political Asylum Seekers" decrying unequal access to education, resettlement and other opportunities notes resignedly:

> The policy of the Kenya Government does not allow asylum seekers to reside in urban areas nor to engage in any productivity. The majority of these asylum seekers are of urban origin and I believe Camp life is not probably ideal for them. They have been declared of concern to UNHCR but [unintelligible] concurrent recognition by the Government of Kenya. To the best of my knowledge it seems that the major resettlement countries are not interested in their cases. Therefore the future situation appears rather bleak.[59]

Another official suggested that perhaps the Nairobi branch office's resettlement unit could review the files of the letter writers, but it is noteworthy that the other concerns in the letter were neither acknowledged nor addressed.

That said, there was the occasional exception. When Elders in Utange mailed a wide-ranging complaint – about inadequate food rations, unavailability of fuel, insufficient water supply, scant medical supplies, and lack of educational opportunities – UNHCR officials in Geneva seemed concerned: "Would think BO Nairobi shld [*sic*] be requested to respond to this letter &, given the seriousness of the problems raised, copy its response to HQs." A subsequent response went further: "Suggest that BO Nairobi visits Utange refugee camp, along with Gvt [*sic*] officials and AMREF [African Medical and Research Foundation, an NGO], to inform the refugees on remedial actions which will be taken." Ultimately, a memo was sent to the branch office for comment.[60]

At any rate, this sort of lengthy commentary by UNHCR officials was the exception rather than the rule. Several letters only received the comment "noted," including the aforementioned document requesting refugee representation at the UN. Most frequently, the action sheets contain terse instructions to forward the letter to the in-country branch office – even in situations where the letter was complaining about mistreatment by, or corruption at, that very same branch office. A typical memo to the Nairobi branch office reads: "Attached please find photocopy of a letter addressed to us by the above-named persons, which we have not acknowledged. We should be grateful if you would take whatever action you deem necessary, and keep us informed ..."[61] An RBA legal advisor seemed taken aback when an RBA desk officer sent him a letter from refugees, writing "This kind of communication should just be addressed/copied to B.O. Kenya for their info."[62] Indeed, when one UNHCR official suggested that High Commissioner Sadako Ogata reply to a letter that had been addressed to her by name, another official responded: "If letter has already been attentioned to BO Kenya for reply/action, I do not think that any further reply is necessary and certainly none by the HC!"[63]

Only letters from third parties, particularly those based in Western countries, seemed to elicit a written reply from UNHCR headquarters. For example, a letter from a group in Ontario, Canada – the Thunder Bay Friends of Refugees – reported a number of complaints received from Ugandan refugees in Nairobi: new asylum seekers were being turned away without processing, recognized refugees' files were not being passed on to the Canadian Consulate for possible resettlement, refugees were refused support allowances and start-up loans even as they were unable to secure legal employment, and UNHCR officials were harassing the refugees and threatening their removal.[64] The comments on the attached sheet note that "It appears to be a serious issue" and record that comments are being sought from the branch office before sending a reply.[65]

Similarly, an October 1990 letter was sent to the High Commissioner from a Geneva-based Comité pour les droits de l'homme et la démocratie au Rwanda (Committee for Human Rights and Democracy in Rwanda), expressing their indignation and astonishment ("indignation et stupeur") that the Kenyan government had instructed police forces to round up Ugandan and Rwandan refugees for forced repatriation. The letter pleaded with the High Commissioner to use his influence in order to reverse Kenya's policies ("d'user de toutes vos compétences et influences pour que ces mesures prises par le Gouvernment Kenyan contre ces réfugiés soient immédiatement levés").[66] In response, the Director of RBA requested that a reply be drafted and that the Regional Bureau for Europe be informed as well.[67] The reply was sent the following month, expressing reassurance that UNHCR had immediately contacted the Kenyan government when the expulsion order was announced and learned that it only concerned undocumented foreigners. The Nairobi branch office would ensure the release of any refugees or asylum seekers who were inadvertently arrested.[68]

Letters from a Kenya-based refugee-led organization elicited a starkly different response. A pair of letters from a refugee community organization head-quartered in Nairobi, the African Refugee Alliance (AFREA), were typed on letterhead with the motto "Help Refugees to Help Themselves" and signed by an Executive Chairman and a Secretary. One letter begins "As a new born in this jurisdiction, we hope the UNHCR will be pleased to nurse, to feed, to clothe and to take care of AFREA until it grows up, so that at its turn, it can work for UNHCR's goals in this motherland Africa."[69] In a letter to UNHCR's branch office in Nairobi and cc'ed to UNHCR headquarters, AFREA reported that their Executive Committee had decided to send three of their members to accompany a UNHCR team that was preparing to visit refugees recently displaced from southern Sudan into Kenya.[70] The attached action sheet contains a single dismissive comment: "Organization not considered appropriate implementing partner by BO Nairobi."[71]

Indeed, the only letter from refugees in Kenya that might have provoked a written response from headquarters was the complaint that a UNHCR staffer was depriving refugees and ought to be expelled. On the attached action sheet, an official from UNHCR headquarters commented indignantly "I hope a reply is being sent to this letter as Ms. Nakano has to be defended from racist allegations and for doing her job properly."[72]

## 5.   Conclusion

Refugees are often portrayed as lacking agency, "like corks bobbing along on the surface of an unstoppable wave of displacement" (Gatrell 2013, 9). Indeed, refugees must navigate a seemingly unending series of power imbalances. They face structural barriers and inequities in their home and host countries, and during the journeys they make. Even UNHCR, the international organization responsible for their protection, wields authority and mobilizes resources that far outpace those available to refugees. The result, as Moulin and Nyers (2007, 361) put it, is that any negotiation by refugees takes place "in a political terrain that has systematically excluded them."

Nonetheless, the research in this article highlights how refugees exercise agency even in the face of formidable structural constraints. To use Lister's (2004) influential taxonomy of agency in contexts of poverty, which has been applied to refugees by Clark-Kazak (2014) and others, refugees in Kenya "got organized" by engaging in collective action to press for change and "got (back) at" by seeking to overturn unjust or inhumane treatment. In emphasizing the danger they faced and appealing to UNHCR's compassion, they may even have engaged in "victimcy" whereby actors use their agency to portray themselves as helpless victims (Utas 2004).

The letters I analyzed reveal refugees performing citizenship by making rights claims of UNHCR. Their shared refugee identity formed a basis for solidarity and advocacy that transcended national divisions. At the same time, their membership and belonging in the refugee community was premised on differentiating themselves from voluntary economic migrants. The grievances refugees expressed and the demands they made constitute them as political agents rather than apolitical subjects. They referred to international refugee law and UNHCR's own statutes in an effort to hold the Agency accountable.

In turn, the Agency largely ignored their appeals. Though it functioned as a "surrogate state," UNHCR operated as a bureaucracy without accountability to its "citizens." From the perspective of the letter writers, whose queries received no response or acknowledgement, it must have seemed as though their missives disappeared into the "black box of bureaucracy" (Thomson 2012).[73] UNHCR's bureaucratic structure and organizational culture led to a rejection of refugees' efforts to constitute themselves as political agents and active participants in the global refugee regime. Here, as elsewhere, bureaucracy produced indifference rather than accountability (Herzfeld 1992). Headquarter staff's comments on the letters even suggest they were seen as an interruption of or a distraction from their work. To quote a UNHCR official, "It's difficult for UNHCR to admit that they don't like dealing with refugees" (qtd. in Wigley 2005, 34).

There is some room for optimism, however. Echoing some of the demands described above, the first ever Global Summit of Refugees was held in 2018, from which emerged the Global Refugee-Led Network (GRN) which seeks refugee participation in local, national, regional, and global decision-making. Recent years have seen suggestions for a UN Declaration on the Participation of Refugees in Decision Making (Harley and Hobbs 2020). It is certainly too soon to consider meaningful refugee participation an established norm (Milner, Alio, and Gardi 2022), but these developments suggest that refugees' acts of citizenship may in fact be altering the refugee regime.

# Acknowledgements

I am grateful to Oliver Bakewell, Saskia Bonjour, Ingrid Eagly, Evelyn Ersanili, Laura Madokoro, Nando Sigona, Darshan Vigneswaran, the participants in the UCLA 2023 International Conference on Forced Migration, and the participants in the *Migration Politics* work-in-progress workshop for providing valuable feedback on earlier drafts of this article. Aditya Srinivasan and Rachelly Buzzi helped digitize documents. Thank you to the staff at the UNHCR Archives in Geneva, Switzerland for their generous assistance with locating and retrieving documents.

# Endnotes

[1] Similarly, Feldman (2008, 10) explains that the UN Relief and Works Agency (UNRWA), "as much as Egypt, governed Gaza" during the period 1950 to 1967 since the majority of the strip's inhabitants were refugees.

[2] While it is possible that intermediaries were involved in transcribing, translating into English, or typing the letters, there is nothing in the documents themselves to suggest that this is the case.

[3] The formation of the Refugee Olympic Athletes Team, starting in 2016, resonates with the notion of a distinct refugee political community.

[4] Representatives of the Multi-Ethnic Ethiopian Refugees' Community at Walda Refugee Camp to UNHCR Headquarters, 8 December 1992, *UNHCR Archives*, Fonds 11, Series 3, 100 KEN ETH Refugee Situations - Special Groups of Refugees - Ethiopian Refugees in Kenya, Folio 15. [Hereafter: Multi-Ethnic Ethiopian Refugees' Community at Walda, 8 December 1992.]

[5] Abdalla Hassan Matuga [Chairman of Uganda Nubian Exiles at Busia, Kenya] to UNHCR Headquarters, 30 May 1983, "Appealing for Assistance and Recognition by the United Nations and Red Cross," *UNHCR Archives*, Fonds 11, Series 2, Box 1198, 630 KEN Protection and General Legal Matters - Eligibility - Kenya [Volume 2], Folio 151. [Hereafter: Abdalla Hassan Matuga, 30 May 1983.]

[6] Somali Women Group - Thika Reception Center to UNHCR Geneva, 18 November 1992, "Assistance," *UNHCR Archives*, Fonds 11, Series 3, 100 KEN SOM Refugee Situations - Special Groups of Refugees – Somalian Refugees in Kenya [Volume B], Folio 21.

[7] Ethiopian Refugee Committee to Representative UNHCR Nairobi, 22 July 1992, *UNHCR Archives*, Fonds 11, Series 3, 100 KEN ETH Refugee Situations - Special Groups of Refugees - Ethiopian Refugees in Kenya, Folio 14.

[8] Committee Members to UNHCR Headquarters, 11 February 1993, *UNHCR Archives*, Fonds 11, Series 3, 100 KEN ETH Refugee Situations - Special Groups of Refugees - Ethiopian Refugees in Kenya, Folio 17.

[9] Committee for Ethiopian Civil & Military Officials to Sadako Ogata, 18 February 1993, *UNHCR Archives*, Fonds 11, Series 3, 100 KEN ETH Refugee Situations - Special Groups of Refugees - Ethiopian Refugees in Kenya, Folio 18. [Hereafter: Ethiopian Civil & Military Officials, 18 February 1993.]

[10] Ethiopian Refugee Community at Ruiru Transit Center to Sadako Ogata, 23 February 1994, *UNHCR Archives*, Fonds 11, Series 3, 100 KEN ETH Refugee Situations - Special Groups of Refugees - Ethiopian Refugees in Kenya, Folio 21. [Hereafter: Ethiopian Refugee Community at Ruiru, 23 February 1994.]

[11] Disabled Committee at Marafa Refugee Camp to UNHCR Geneva, 11 April 1994, "To Request Resettlement and Better Treatment for Physically and Mentally Disabled in Jes-Maraf Refugee Camp," *UNHCR Archives*, Fonds 11, Series 3, 100 KEN GEN Refugee Situations - Special Groups of Refugees - Refugees in Kenya [Volume B], Folio 21.

[12] Refugees in Thika Refugee Camp to UNHCR Headquarters, "SOS," *UNHCR Archives*, Fonds 11, Series 3, 100 KEN GEN Refugee Situations - Special Groups of Refugees - Refugees in Kenya [Volume A], Folio 8. [Hereafter: Refugees in Thika, "SOS."]

[13] Joint Ethio-Somali Committee of Refugees in Thika to UNHCR Geneva, 13 March 1992, "Joint Communique of the Ethio-Somali Refugees at Thika R.C.R. Declared against the Issue of Transfer to Walda, Liboi, and Ifo Refugee Camps," *UNHCR Archives*, Fonds 11, Series 3, 100 KEN SOM Refugee Situations - Special Groups of Refugees – Somalian Refugees in Kenya [Volume B], Folio 18. [Hereafter: Joint Ethio-Somali Committee in Thika, 13 March 1992.]

[14] The only exception was a letter which contained a general description of conditions in Ethiopia: "serious economic crisis" was listed fourth, after human rights violations, civil war, and fanatic leadership. Ethiopian Refugees in Marsabit to UNHCR Geneva, 18 December 1990, *UNHCR Archives*, Fonds 11, Series 3, 100 KEN ETH Refugee Situations - Special Groups of Refugees - Ethiopian Refugees in Kenya, Folio 6. [Hereafter: Ethiopian Refugees in Marsabit, 18 December 1990.]

[15] Joint Ethio-Somali Committee in Thika, 13 March 1992.

[16] Ethiopian Refugees at Walda to UNHCR Geneva, 5 October 1991, *UNHCR Archives*, Fonds 11, Series 3, 100 KEN ETH Refugee Situations - Special Groups of Refugees - Ethiopian Refugees in Kenya, Folio 10. [Hereafter: Ethiopian Refugees at Walda, 5 October 1991.]

[17] The only exception is a letter that refers to "the merciless and terroric [*sic*] inter-clans exploded civil war in Somalia starting from 1991." Utange Refugees Camp Complainant Elders to UNHCR Headquarters, 1 October 1992, "Claim and Complain," *UNHCR Archives*, Fonds 11, Series 3, 100 KEN SOM Refugee Situations - Special Groups of Refugees – Somalian Refugees in Kenya [Volume B], Folio 24. [Hereafter: Utange Complainant Elders, 1 October 1992.]

[18] Abdalla Hassan Matuga, 30 May 1983.

[19] Ibid.

[20] Ibid.

[21] Joint Ethio-Somali Committee in Thika, 13 March 1992.

[22] Elders of Utange Refugee Camp to UNHCR Nairobi, 11 September 1991, "Appeal for Intervention," *UNHCR Archives*, Fonds 11, Series 3, 100 KEN GEN Refugee Situations - Special Groups of Refugees - Refugees in Kenya [Volume A], Folio 12. [Hereafter: Elders of Utange, 11 September 1991.]

[23] Ethiopian Refugees Community at Kakuma Refugee Camp, June 1994, UNHCR Archives, Fonds 11, Series 3, 100 KEN SOM Refugee Situations - Special Groups of Refugees – Somalian Refugees in Kenya [Volume B], Folio 26. [Hereafter: Ethiopian Refugees at Kakuma, June 1994.]

[24] Multi-Ethnic Ethiopian Refugees' Community at Walda, 8 December 1992.

[25] Ibid.

[26] Elders of Majerteen, Meheri, and Kasiqabe and Madhiban to UNHCR Sub-Office Mombasa, 28 December 1993, "A Protest Against the Newly Formed Elders Committee of Utange R. Camp," *UNHCR Archives*, Fonds 11, Series 3, 100 KEN GEN Refugee Situations - Special Groups of Refugees - Refugees in Kenya [Volume A].

[27] The Elder's Council of Utange Refugee Camp to UNHCR Geneva, 3 March 1992, "Appeal for Protection," *UNHCR Archives*, Fonds 11, Series 3, 100 KEN SOM Refugee Situations - Special Groups of Refugees – Somalian Refugees in Kenya [Volume B], Folio 17A.

[28] ? to Under-Secretary MHA, 13 June 1986, "Assistance," *UNHCR Archives*, Fonds 11, Series 3, 100 KEN GEN Refugee Situations - Special Groups of Refugees - Refugees in Kenya [Volume A], Folio 6.

[29] ? to UNHCR Geneva, 21 March 1991, "Denial of Refugees Rights of Ugandan Refugees Rights by the UNHCR Kenya," *UNHCR Archives*, Fonds 11, Series 3, 100 KEN UGA Refugee Situations - Special Groups of Refugees – Ugandan Refugees in Kenya, Folio 11.

[30] Ethiopian refugees at Thika to Sadako Ogata, 20 December 1991, "The Problem of Protection and the UNHCR Branch Office of Kenya," *UNHCR Archives*, Fonds 11, Series 3, 100 KEN ETH Refugee Situations - Special Groups of Refugees - Ethiopian Refugees in Kenya, Folio 11. [Hereafter: Ethiopian refugees at Thika, 20 December 1991.]

[31] Mathew Kibiramgo to UNHCR Geneva, 21 October 1990, "Save Us from this Influx," *UNHCR Archives*, Fonds 11, Series 3, 100 KEN GEN Refugee Situations - Special Groups of Refugees - Refugees in Kenya [Volume A], Folio 9.

[32] Ethiopian Refugees at Walda, 5 October 1991.

[33] Ethiopian refugees at Thika, 20 December 1991.

[34] Elders of Utange, 11 September 1991.

[35] ? to UN Secretary-General, 22 November 1990, "The Human Rights of Refugees," *UNHCR Archives*, Fonds 11, Series 3, 100 KEN GEN Refugee Situations - Special Groups of Refugees - Refugees in Kenya [Volume A], Folio 10.

[36] ? to UN Secretary-General, 22 November 1990, "The Human Rights of Refugees," *UNHCR Archives*, Fonds 11, Series 3, 100 KEN GEN Refugee Situations - Special Groups of Refugees - Refugees in Kenya [Volume A], Folio 10.

[37] Ibid.

[38] Refugees in Thika, "SOS."

[39] Charles Peter Karekesi and Others to Minister of Home Affairs, 21 July 1987, "Kenya Refugees Problem," *UNHCR Archives*, Fonds 11, Series 3, 100 KEN GEN Refugee Situations - Special Groups of Refugees - Refugees in Kenya [Volume A], Folio 7. [Hereafter: Charles Peter Karekesi, 21 July 1987.]

[40] Ethiopian Civil & Military Officials, 18 February 1993.

[41] Ethiopian Refugees in Marsabit, 18 December 1990.

[42] Abdalla Hassan Matuga, 30 May 1983.

[43] Utange Complainant Elders, 1 October 1992.

[44] Ethiopian Refugee Community at Ruiru, 23 February 1994.

[45] Ethiopian Refugees at Walda, 5 October 1991.

[46] For example, Ethiopian Civil & Military Officials, 18 February 1993.

[47] Ethiopian Refugees at Kakuma, June 1994.

[48] Kedir Wabella Salih [Representative of Ethiopian Refugees in Kenya] to UNHCR Geneva, 11 June 1987, *UNHCR Archives*, Fonds 11, Series 3, 100 KEN ETH Refugee Situations - Special Groups of Refugees - Ethiopian Refugees in Kenya, Folio 2.

[49] Kedir Wabella Salih [Representative of Ethiopian Refugees in Kenya] to UNHCR Geneva, 17 June 1987, *UNHCR Archives*, Fonds 11, Series 3, 100 KEN ETH Refugee Situations - Special Groups of Refugees - Ethiopian Refugees in Kenya, Folio 3.

[50] Kedir Wabella Salih [Representative of Ethiopian Refugees in Kenya] to UNHCR Geneva, 1 July 1987, *UNHCR Archives*, Fonds 11, Series 3, 100 KEN ETH Refugee Situations - Special Groups of Refugees - Ethiopian Refugees in Kenya, Folio 4.

[51] Richard Okwon Wod Obal [Secretary TRCR Committee] to UNHCR Nairobi, 19 March 1990, "Ugandans Position of Interviews," *UNHCR Archives*, Fonds 11, Series 3, 100 KEN UGA Refugee Situations - Special Groups of Refugees – Ugandan Refugees in Kenya, Folio 8.

[52] Charles Peter Karekesi, 21 July 1987.

[53] Ex-Somali Refugees to UNHCR Geneva, 20 June 1991, "Grievance Letter," *UNHCR Archives*, Fonds 11, Series 3, 100 KEN SOM Refugee Situations - Special Groups of Refugees – Somalian Refugees in Kenya [Volume A], Folio 10.

[54] Ethiopian Refugees in Marsabit, 18 December 1990.

[55] Ibid.

[56] Utange Refugee Community to Provincial Commissioner Coast Province Mombasa, 12 June 1992, "Appeal for Protection against Burning Threat made by Mr. Bamahriz," *UNHCR Archives*, Fonds 11, Series 3, 10 KEN External Relations - Relations with Governments - Kenya [Volume B], Folio 20.

[57] Action Sheet, 18 March 1992, *UNHCR Archives*, Fonds 11, Series 3, 100 KEN SOM Refugee Situations - Special Groups of Refugees – Somalian Refugees in Kenya [Volume B], Folio 18.

[58] Peter Meijer, Head of Desk II, Regional Bureau for Africa to Carrol Faubert [Representative BO Kenya], 9 April 1992, "Closure of Thika – Protest by Refugees," *UNHCR Archives*, Fonds 11, Series 3, 100 KEN SOM Refugee Situations - Special Groups of Refugees – Somalian Refugees in Kenya [Volume B], Folio 21.

[59] Action Sheet, 3 April 1994, *UNHCR Archives*, Fonds 11, Series 3, 100 KEN ETH Refugee Situations - Special Groups of Refugees - Ethiopian Refugees in Kenya, Folio 21.

[60] Action Sheet, 4 October 1991, *UNHCR Archives*, Fonds 11, Series 3, 100 KEN GEN Refugee Situations - Special Groups of Refugees - Refugees in Kenya [Volume A], Folio 12.

[61] Peter Meijer, Head of Desk II, Regional Bureau for Africa to Representative, UNHCR Branch Office in Kenya, 7 February 1992, UNHCR Archives, Fonds 11, Series 3, 100 KEN ETH Refugee Situations - Special Groups of Refugees - Ethiopian Refugees in Kenya, Folio 13.

[62] Action Sheet, 20 June 1994, UNHCR Archives, Fonds 11, Series 3, 100 KEN SOM Refugee Situations - Special Groups of Refugees – Somalian Refugees in Kenya [Volume B], Folio 26.

[63] Action Sheet, 24 February 1993, *UNHCR Archives*, Fonds 11, Series 3, 100 KEN ETH Refugee Situations - Special Groups of Refugees - Ethiopian Refugees in Kenya, Folio 18.

[64] Liz McWeeny [Thunder Bay Friends of Refugees] to UNHCR Geneva, 24 October 1990," *UNHCR Archives*, Fonds 11, Series 3, 100 KEN UGA Refugee Situations - Special Groups of Refugees – Ugandan Refugees in Kenya, Folio 10.

[65] Action Sheet, 6 November 1990," *UNHCR Archives*, Fonds 11, Series 3, 100 KEN UGA Refugee Situations - Special Groups of Refugees – Ugandan Refugees in Kenya, Folio 10.

[66] Pierre Karemera [Comité pour les droits de l'homme et la démocratie au Rwanda] to UN High Commissioner for Refugees, 23 October 1990, *UNHCR Archives*, Fonds 11, Series 3, 100 KEN RWA Refugee Situations - Special Groups of Refugees – Rwandan Refugees in Kenya, Folio 1.

[67] Action Sheet, 26 October 1990, *UNHCR Archives*, Fonds 11, Series 3, 100 KEN RWA Refugee Situations - Special Groups of Refugees – Rwandan Refugees in Kenya, Folio 1.

[68] N. Bwakira [Director, Regional Bureau for Africa] to Pierre Karemera, 21 November 1990, *UNHCR Archives*, Fonds 11, Series 3, 100 KEN RWA Refugee Situations - Special Groups of Refugees – Rwandan Refugees in Kenya, Folio 2.

[69] African Refugee Alliance to UNHCR Utange, 5 June 1992, "Introduction of AFREA's Activists," *UNHCR Archives*, Fonds 11, Series 3, 100 KEN GEN Refugee Situations - Special Groups of Refugees - Refugees in Kenya [Volume A], Folio 16.

[70] African Refugee Alliance to UNHCR Nairobi, 2 June 1992, "Three Members From AFREA to Be Part of Your Team to Lokichogio and Gakum Camps," *UNHCR Archives*, Fonds 11, Series 3, 100 KEN GEN Refugee Situations - Special Groups of Refugees - Refugees in Kenya [Volume A], Folio 16.

[71] Action Sheet, 25 June 1992, *UNHCR Archives*, Fonds 11, Series 3, 100 KEN GEN Refugee Situations - Special Groups of Refugees - Refugees in Kenya [Volume A], Folio 16.

[72] Action Sheet, 11 August 1987, *UNHCR Archives*, Fonds 11, Series 3, 100 KEN GEN Refugee Situations - Special Groups of Refugees - Refugees in Kenya [Volume A], Folio 7.

[73] Then again, Malkki (1995, 151) suggests that Burundian Hutu refugees sent letters from Mishamo Refugee Settlement in Tanzania to international organizations in Europe, not in expectation of receiving a reply, but to document their situation in detail: "These letters seemed to be meant as a constant reminder to the world: We are still refugees and still under your mandate."

# Appendix

I coded text segments in each document based on the following set of dimensions:

- For each refugee letter:
  - o Self-identification: What collective identity do the authors/signatories use to describe themselves? (e.g., national identity, camp residence, former occupation, gender)
  - o Displacement narrative: What reason or account of events is recounted to explain and describe their flight? (e.g., persecution, violence, starvation)
  - o Complaints: Who and what do refugees find fault with? (e.g., camp conditions, local UNHCR personnel, government officials)
  - o Requests: What do refugees call for as a response to their complaint? (e.g., supplies, repatriation, resettlement)
  - o Argumentation: What justifications do refugees advance to obtain UNHCR's help? (e.g., invoking their rights)
  - o Emphases: Which words or phrases are underlined or written in capital letters? What sections and subsections is the letter divided into?
  - o Tone: What is the tenor of the language in the letter? (e.g., deferential, indignant, assertive)
  - o Format: Who is the letter addressed to? What salutation is used? How is it signed and by whom? Which, if any, other organizations or individuals are cced?
  - o Prior correspondence: Are there references to previous letters sent?

- For the UNHCR action sheet and inter-office correspondence
  - o Job titles: Which officials respond and what positions do they hold within the UNHCR hierarchy?
  - o Response: What action is recommended or undertaken (if any)? (e.g., acknowledge letter, no action taken)
  - o Argumentation: What justifications do officials advance to support their response? (e.g., refer to resource constraints)
  - o Emphases: Which words or phrases are underlined or written in capital letters?
  - o Tone: What is the overall tenor of the language in the documentation? (e.g., dismissive, concerned)
  - o Routing: To whom is the letter forwarded? (e.g., in-country branch office)

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
