# Peer review of ""The Greatest and Most Important Human Right": Citizenship and Bureaucratic Indifference in Refugee-UNHCR Correspondence"

_Migration Politics_

## Round 1 · Author Response

Thank you to the three reviewers for their comments, and to the editor for the opportunity to revise my article manuscript.

I have addressed the changes requested and revised the paper accordingly. Below, I number each requested change in quotation marks and place my response below it.

Thank you for your time and consideration.

---

## Round 1 · List of Changes

REVIEWER 2 COMMENTS

  1. “In the opening section the author states that the article "puts into conversation three bodies of literature." I recommend enumerating these three bodies more explicitly, as it is a little unclear at present.”

  2. Author’s response: Thank you for this point. This sentence now lists “citizenship studies, scholarship on refugee petitions, and research on international organizations.”

  3. “I would suggest bringing in some of Ilana Feldman's work on refugee bureaucracy in the Gaza context, which seems highly relevant here.”

  4. Author’s response: I appreciate this suggestion and read the book with interest. Feldman’s discussion focuses on UNRWA rather than UNHCR, and there are of course differences between the two organizations and the populations they work with – as well as differences between her time period (1917-1967) and my own. So, I mention her work briefly in endnote 1.

  5. “The article could engage slightly more with conceptualising citizenship and the relevant literature on this.”

  6. Author’s response: I am not sure whether the reviewer has specific pieces in mind. My discussion of citizenship and the citations that I include focus on those aspects of citizenship and its conceptualization that are most relevant to my analysis: performative citizenship (Isin 2017), acts of citizenship in the absence of legal status (Swerts 2017), membership and boundary-drawing (Cooper 2016; Isin 2008), implicit acceptance of governance and authority (Leonardi & Vaughan 2016), and acts of citizenship as a rejection of perceptions of passivity (Nyers 2008).

REVIEWER 3 COMMENTS

  1. “The paragraph in the intro that describes the contributions to literatures was a bit convoluted (and in passive voice), it would be good to state the three literatures up front because I read it several times and couldn’t quite segment them out (ex. “This article puts three bodies of literature into conversation: citizenship studies, international organizations, and ?). A few citations in this par to preview the lit review would also be helpful. As I read through the article, I thought this speaks to different aspects of citizenship studies lit rather than three different lits (but it’s the author’s determination). It just needs to be made a bit clearer and more explicit from the get go.”

  2. Author’s response: Thank you for pointing this out. The paragraph now clarifies that the three literatures are “citizenship studies, scholarship on refugee petitions, and research on international organizations.” I have also added a few example citations as requested, but leave the bulk to be accompanied by more extended and detailed description in the following section.

  3. “I thought the discussion of enacting citizenship as part of the UNHCR as an entity that replaces the state was fascinating (pg 3). It’s not mentioned here, but the fact that the UNHCR has its own Olympic team only adds to the argument.”

  4. Author’s response: What a great observation! The team does not technically represent UNHCR, but I do think this might speak to the idea of refugees constituting their own political community. Since it is a more recent development than the events discussed in the paper, I mention it briefly in endnote 3.

---

## Editorial Decision

unknown